# XAL: EXplainable Active Learning Makes Classifiers Better Low-resource Learners

## ABSTRACT

Active learning aims to construct an effective training set by iteratively curating the most informative unlabeled data for annotation, which is practical in low-resource tasks. Most active learning techniques in classification rely on the model's uncertainty or disagreement to choose unlabeled data. However, previous work indicates that existing models are poor at quantifying predictive uncertainty, which can lead to over-confidence in superficial patterns and a lack of exploration. Inspired by the cognitive processes in which humans deduce and predict through causal information, we propose a novel Explainable Active Learning framework (XAL) for low-resource text classification, which aims to encourage classifiers to justify their inferences and delve into unlabeled data for which they cannot provide reasonable explanations. Specifically, besides using a pre-trained bi-directional encoder for classification, we employ a pre-trained uni-directional decoder to generate and score the explanation. A ranking loss is proposed to enhance the decoder's capability in scoring explanations. During the selection of unlabeled data, we combine the predictive uncertainty of the encoder and the explanation score of the decoder to acquire informative data for annotation.

As XAL is a general framework for text classification, we test our methods on six different classification tasks. Extensive experiments show that XAL achieves substantial improvement on all six tasks over previous AL methods. Ablation studies demonstrate the effectiveness of each component, and human evaluation shows that the model trained in XAL performs surprisingly well well in explaining its prediction.

## 1 INTRODUCTION

Active learning (AL) is a machine-learning paradigm that efficiently acquires data for annotation from a (typically large) unlabeled data pool and iteratively trains models (Lewis & Catlett, 1994; Margatina et al., 2021). AL frameworks have attracted considerable attention from researchers due to their high realistic values reduce the data annotation costs by concentrating the human labeling effort on the most informative data points, which can be applied in low-resources tasks (Lewis & Catlett, 1994; Settles, 2009; Zhang et al., 2022b).

Most previous AL methods rely on model predictive uncertainty or disagreement for the unlabeled data, and the most uncertain data are believed to be the most informative and worthful ones to be annotated (Lewis, 1995; Houlsby et al., 2011; Margatina et al., 2021; Zhang et al., 2022a). However, previous studies have indicated that existing models struggle to accurately quantify predictive uncertainty (Guo et al., 2017; Lakshminarayanan et al., 2017), leading to overconfidence and insufficient exploration, i.e., models tend to choose data instances that are uncertain yet repetitively uninformative (Margatina et al., 2021). This issue arises because training can lead cross-entropy-based classifiers to learn superficial or spurious patterns (Guo et al., 2022; 2023; Srivastava et al., 2020), rather than the causal information between inputs and labels.

In the context of cognitive science and psychological science, humans make decisions or inferences by exploring causal information (Frye et al., 1996; Joyce, 1999; Rottman & Hastie, 2014). For example, when learning to differentiate animals, humans do not merely rely on statistical features such as colors or feathers. They also consider the creatures' habits, such as dietary patterns, and kinship, such as the species of the parents, to engage in causal reasoning, thereby determining the species of

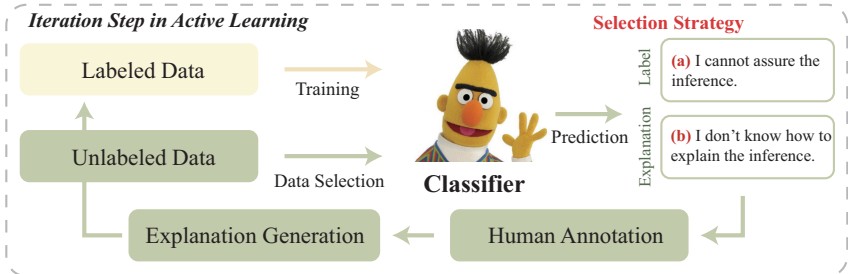

Figure 1: Data selection strategy in Explainable Active Learning (XAL). Previous work selects the unlabeled data mostly relying on the model's uncertainty (a), but XAL proposes to further leverage the model's explanation of its prediction (b).

the organism. Explaining the reasons behind the classification also enhances the justification of the inference confidence. Intuitively, explanations of the causal information can help the model confirm whether it understands how to make classifications. It motivates us to encourage classifiers to learn the causal information behind inferences and to explore unlabeled data for which the model cannot provide reasonable explanations. In doing so, the model can learn the causal relationships between labels and texts and reduce reliance on superficial patterns, which leads to improved generalization and more effective exploration within AL frameworks. The main intuition is illustrated in Figure 1.

Given the above observations, we introduce an Explainable Active Learning Framework (XAL) for text classification tasks. This framework consists of two main components: the training process and the data selection process. Primarily, we adopt a pre-trained bi-directional encoder for classification and a pre-trained uni-directional decoder to generate and score explanations that serve as expressions of causal information in human language. In the training phase, we use the classification labels and explanations to optimize the model parameters. Besides, to further enhance the decoder's ability to score explanations, we design a ranking loss that optimizes the model to differentiate between reasonable and unreasonable explanations. To implement this ranking loss, we generate a variety of explanations (both reasonable and unreasonable) for labeled data by querying ChatGPT with different prompts, thereby eliminating the need for additional human annotation effort. Subsequently, during the data selection phase, we amalgamate the predictive uncertainty of the encoder and the explanation score of the decoder to rank unlabeled data. The most informative data are then annotated and incorporated into further training.

We conduct experiments on various text classification tasks, including natural language inference, paraphrase detection, category sentiment classification, stance detection, (dis)agreement detection, and relevance classification. Experimental results manifest that our model can achieve substantial improvement in all six tasks. Ablation studies demonstrate the effectiveness of each component, and human evaluation shows that the model trained in XAL works well in explaining its prediction. To our knowledge, we are the first to incorporate the model's explanation (explanation score) into evaluating the informativeness of unlabeled data in an AL framework. We will release our code to facilitate future research.

## 2 METHOD

### 2.1 OVERVIEW

**Task Formulation** We mainly consider a $C$ class text classification task defined on a compact set $\mathcal{X}$ and a label space $\mathcal{Y} = \{1, ..., C\}$. The data points are sampled i.i.d over the space $\mathcal{Z} = \mathcal{X} \times \mathcal{Y}$ as $\{\mathbf{x}_i, y_i\} \sim p_z$, which can be divided into two sets – the labeled set $D_l$ and the unlabeled set $D_u$. At the beginning of an active learning algorithm, only a small number of data points are randomly selected into the labeled set $D_l$ and we have only access to data points in $D_l$ for training the classification model. Then $L$ data from $D_u$ are selected for annotation and added to $D_l$ (removed from $D_u$ simultaneously) in $\mathcal{M}$ multiple rounds.

**Model Architecture** Following previous work (Devlin et al., 2018), we adopt a pre-trained bi-directional encoder as the backbone for classification. In addition to the encoder, a corresponding uni-directional decoder is applied to generate and score the explanation for the label prediction. Dur-

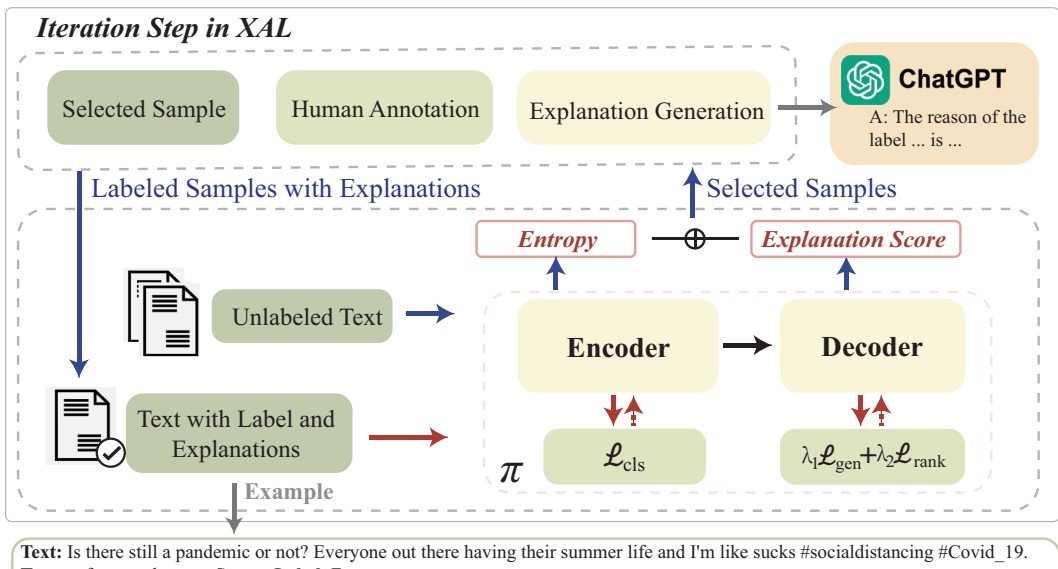

**Text:** Is there still a pandemic or not? Everyone out there having their summer life and I'm like sucks #socialdistancing #Covid_19.
**Target:** face mask          **Stance Label**: Favor
**Explanations:** (1)The stance label is Favor because the user mentions #socialdistancing and #Covid_19... (2)... (3)...

Figure 2: Our proposed XAL framework, which can be divided into two main parts – the training process (red arrows) and the data selection process (blue arrows). The training process aims to train the encoder-decoder model to learn classification and explanation generation. The data selection process aims to select unlabeled data using predictive entropy and explanation scores.

ing training, we construct $k$ different explanations $\mathbf{e}_r$, i.e., $\{\mathbf{e}_r\}_i$, $r = 0, ..., k-1$, for each example $\{\mathbf{x}_i, y_i\}$, where $\mathbf{e}_0$ is the reasonable explanation and $\{\mathbf{e}_{r>0}\}$ are $k-1$ unreasonable explanations. We leave the construction process of explanations in Section 2.4 for further descriptions. Before that, we will first present the model training and data selection in Section 2.2 and Section 2.3 respectively. The framework of the proposed XAL is shown in Figure 2 and the workflow can be found in Algorithm 1.

## 2.2 TRAINING

For each text input $\mathbf{x}$ (we omit all the subscripts of $i$ for simplicity in this subsection), we first prepend it with a special token [CLS] and then obtain the contextual representation by feeding it into the encoder. The contextual representation of the $j$th token is calculated as:

$$\mathbf{h}_j = Encoder([\text{CLS}] + \mathbf{x})[j]. \tag{1}$$

The representation for [CLS], i.e., $\mathbf{h}_0$ is taken as the sentence representation and fed into the classification layer, which is composed of a linear layer and a softmax function. The probability distribution on label space $\mathcal{Y}$ can be formulated as:

$$P(y|\mathbf{x}) = Softmax(Linear(\mathbf{h}_0)). \tag{2}$$

The cross-entropy loss is adopted to optimize the encoder parameters:

$$\mathcal{L}_{cls} = -\sum P(y|\mathbf{x}) \, log \, P(y|\mathbf{x}). \tag{3}$$

Additionally, on the decoder side, the model is trained with teacher forcing to generate the golden explanation $\mathbf{e}_0$. The generation loss is calculated as:

$$\mathcal{L}_{gen} = -\sum_t log \, P(\mathbf{e}_{0,t}|\mathbf{h}, \mathbf{e}_{0,<t}). \tag{4}$$

To make the decoder a good scorer to rank the reasonable and unreasonable explanations, we additionally adopt a ranking loss to optimize the decoder. In particular, the model is trained to rank between reasonable and unreasonable explanations. The ranking loss can be formulated as:

$$\mathcal{L}_{rank} = \sum_{r>0} max(0, p_r - p_0), \tag{5}$$

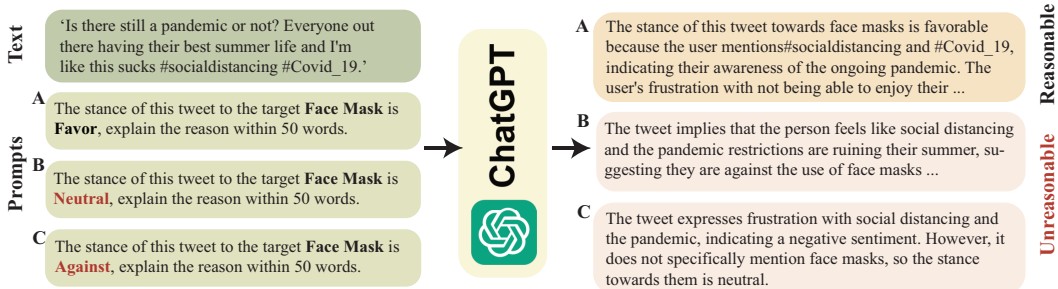

Figure 3: The process to generate diverse explanations from LLMs. We can obtain reasonable and unreasonable explanations by querying ChatGPT with correct and incorrect labels, respectively.

where $p_r$ is the explanation score for $\mathbf{e}_r$, which is calculated as the length-normalized conditional log probability:

$$p_r = \frac{\sum_t logP(\mathbf{e}_{r,t}|\mathbf{x}, \mathbf{e}_{r,<t})}{||\mathbf{e}_r||}. \tag{6}$$

The hyper-parameters are adopted to balance the weights of each loss, and the overall loss is formalized as follows:

$$\mathcal{L} = \mathcal{L}_{cls} + \lambda_1 \mathcal{L}_{gen} + \lambda_2 \mathcal{L}_{rank}. \tag{7}$$

## 2.3 DATA SELECTION

After training the model in each iteration, we can obtain an intermediate model $\pi$. To select the informative data in the unlabeled set $\mathcal{D}_u$, we adopt a combination of the predictive entropy and explanation score. Specifically, for each raw data $\mathbf{x}_i \in \mathcal{D}_u$, we first generate the explanation $\mathbf{e}_i$ by selecting the top-1 output in the beam search. Then, we calculate the explanation score $p_i$ as Eq. 6 and the predictive entropy $c_i$ as Eq. 3. The final score $s_i$ for example $\mathbf{x}_i$ is calculated as the weighted sum of the normalized explanation score and predictive entropy:

$$s_i = \frac{\lambda}{1+\lambda} \frac{e^{-p_i}}{\sum_i e^{-p_i}} + \frac{1}{1+\lambda} \frac{e^{c_i}}{\sum_i e^{c_i}} \tag{8}$$

where the $\lambda$ is the hyper-parameter to balance the explanation score and the predictive entropy. With the final score for each example, we rank the whole unlabeled instances and select the top $L$ instances for annotation.

## 2.4 GENERATION OF GOLDEN EXPLANATIONS

With the advancement of LLMs, previous work has shown that LLMs are good at reasoning (Bang et al., 2023; Rajasekharan et al., 2023). Inspired by these studies, we take the LLMs, such as Chat-GPT and GPT4, as the teacher models, and query them to generate explanations for each selected labeled data, eliminating the annotation cost of human labor. In particular, we design slightly different prompt templates for different tasks, and the prompt for each task is shown in Appendix A. Taking stance detection as an example, its prompt template is designed as *'The stance of this tweet to the target {**Target**} is {**Label**}, explain the reason within 50 words'*, where the **Target** is the corresponding stance target, and the **Label** is the classification label. The final query to the teacher model is the concatenation of the text and the prompt. We construct a reasonable explanation by feeding the golden label into the query and generate several unreasonable explanations by feeding wrong labels. Figure 3 shows an example that we generate explanations by querying ChatGPT, where we can observe that ChatGPT could provide different explanations according to the label we offer it.

## 3 EXPERIMENTS

## 3.1 IMPLEMENTATION DETAILS

In our experiments, we directly utilize a pre-trained encoder-decoder language model for its strong ability in text understanding and generation. Specifically, we adopt the officially released pre-trained

| Task | Dataset | # Labels | Train | Dev | Test |
|------|---------|----------|-------|-----|------|
| Natural Language Inference | RTE (Bentivogli et al., 2009) | 2 | 2,240 | 250 | 278 |
| Paraphrase Detection | MRPC (Dolan et al., 2004) | 2 | 3,667 | 409 | 1,726 |
| Stance Detection | COVID19 (Glandt et al., 2021) | 3 | 4,533 | 800 | 800 |
| Category Sentiment Classification | MAMS (Jiang et al., 2019) | 3 | 7,090 | 888 | 901 |
| (Dis)agreement Detection | DEBA (Pougué-Biyong et al., 2021) | 3 | 4,617 | 578 | 580 |
| Relevance Classification | CLEF (Kanoulas et al., 2017) | 2 | 7,847 | 981 | 982 |

Table 1: All the six text classification tasks used in our experiments.

FLAN-T5-Large model (Chung et al., 2022) from Huggingface [1]. All models in our experiments are trained on a single GPU (Tesla V100) using the Adam optimizer (Kingma & Ba, 2014). We set the learning rate at 1e-4, with a linear scheduler. The batch size is consistently set to 1 across all tasks. The models are trained for 10 epochs in each iteration. Hyper-parameters $\lambda_1$, $\lambda_2$, and $\lambda$ are set to 0.1, 0.01, and 0.5, respectively, based on preliminary experiments. The performance for all tasks is evaluated based on macro-averaged F1. The reported results are the average of three initial sets $D_l$ and three random seeds.

## 3.2 TASKS AND DATASET

We conduct experiments on six different text classification tasks and the data statistics for each task are shown in Table 1 [2]: (1) **Natural Language Inference** aims to detect whether the meaning of one text is entailed (can be inferred) from the other text; (2) **Paraphrase Detection** requires identifying whether each sequence pair is paraphrased; (3) **Category Sentiment Classification** aims to identify the sentiment (Positive/Negative/Neutral) of a given review to a category of the target such as food and staff; (4) **Stance Detection** aims to identify the stance (Favor/Against/Neutral) of a given text to a target; (5) **(Dis)agreement Detection** aims to detect the stance (Agree/Disagree/Neutral) of one reply to a comment; (6) **Relevance Classification** aims to detect whether a scientific document is relevant to a given topic. Appendix A demonstrates the details of six datasets with examples.

## 3.3 BASELINES

To demonstrate the effectiveness of our proposed method, we compare XAL with the following seven AL baselines: (1) **Random** uniformly selects unlabeled data for annotation; (2) **Max-Entropy (ME)** (Lewis, 1995; Schohn & Cohn, 2000) calculates the predictive entropy in the current model and selects data with max entropy ; (3) **Bayesian Active Learning by Disagreement (BALD)** (Houlsby et al., 2011) exploits the uncertainty of unlabeled data by applying different dropouts at test time; (4) **Breaking Ties (BK)** (Scheffer et al., 2001) selects instances with the minimum margin between the top two most likely probabilities ; (5) **Least Confidence (LC)** (Culotta & McCallum, 2005) adopts instances whose most likely label has the least predictive confidence; (6) **Coreset** (Sener & Savarese, 2018; Chai et al., 2023) treats the data representations in the labeled pool $D_u$ as cluster centers, and the unlabeled data with the most significant L2 distance from its nearest cluster center are selected for annotation; (7) **Contrastive Active Learning (CAL)** (Margatina et al., 2021) selects instances with the maximum mean Kullback-Leibler (KL) divergence between the probability distributions of an instance and its $m$ nearest neighbors.

## 4 RESULTS AND DISCUSSION

## 4.1 MAIN RESULTS

We mainly consider two different settings: (1) Given the data selection budget, we observe the trend of changes in model performance; (2) Given the performance upper bound, we observe the number of required instances that the model needs to achieve 90% of the upper-bound performance.

---

[1] https://huggingface.co/

[2] Without losing generality, we randomly split the training set in RTE, and MRPC into train/dev set with proportion 9:1. In DEBA, we adopt the topic of climate change for experiments. The selected data and generated explanations will be released for reproduction.

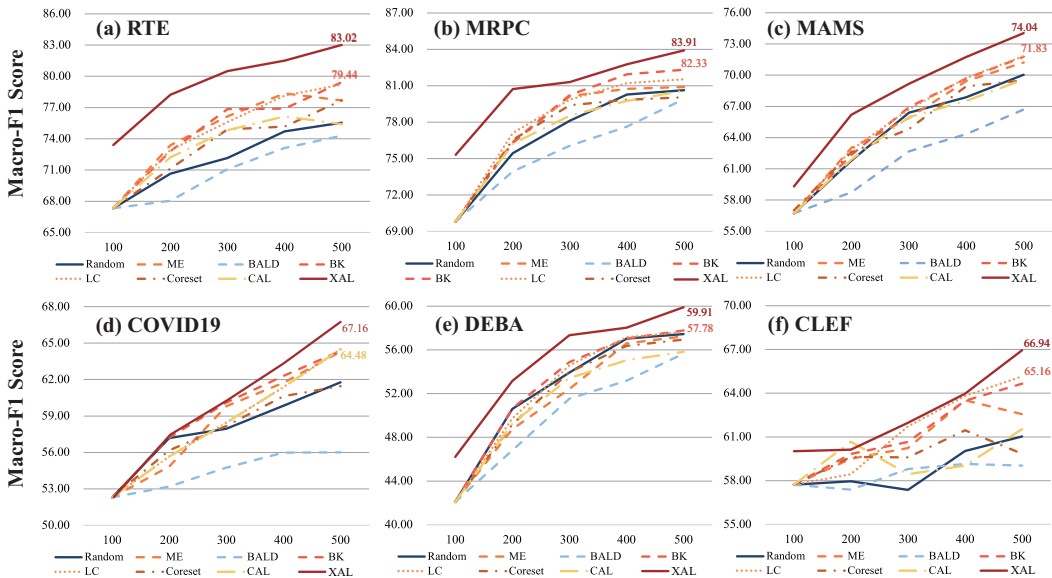

Figure 4: Results given the data selection budget 500 instances in six text classification tasks, where 100 instances are selected for annotation in each iteration. Here we plot the specific values of XAL and the second significant performance when using 500 instances, and the detailed performance values can be found in Appendix D.1.

**Given Data Selection Budget** Following previous work (Zhang et al., 2017; Schröder et al., 2022), we set the data selection budget as 500 instances and select 100 instances for annotation in each iteration. The results are presented in Figure 4. We can observe that the proposed XAL model consistently outperforms other active learning methods. For instance, in the RTE task, our model attains a macro-F1 score of 83.02% at the end, which is 3.58% higher than the second-best result (BK at 79.44%). Similarly, in the DEBA task, XAL surpasses the second-best result (BK at 57.78%) by 2.13%. These results demonstrate the effectiveness of our XAL framework in addressing text classification tasks. We also evaluate ChatGPT on these datasets in the zero-shot setting, where the macro-F1 scores on test sets are 77.42%, 72.46%, 61.28%, 66.67%, 48.96% and 46.41% for RTE, MRPC, MAMS, COVID19, DEBA, and CLEF, respectively. The models with supervised fine-tuning of less than 500 labeled data in XAL significantly outperform ChatGPT on these datasets, which indicates that our model can achieve satisfactory performance with low resources and cost.

In the stance detection task, while the model does not significantly outperform the baselines at the beginning (possibly due to the relatively high complexity of the task), it still exhibits stronger performance with a data count of 300-500, which underscores the effectiveness of the data selection in XAL. In the CLEF dataset, we notice that the performance of baseline models is notably unstable due to the significant imbalance in label distribution (the ratio between relevant and irrelevant is approximately 1:21). However, our XAL model achieves superior performance and more consistent improvements over baselines during the data selection process, which validates the effectiveness of our model structure and data selection strategy, even in challenging scenarios of imbalanced data.

**Given Performance Upper Bound** It's also valuable to evaluate the amount of data required for models to achieve comparable performance with those trained on the entire training dataset. Specifically, we begin with an initial labeled set of 100 instances and select a certain number of instances to annotate in each selection iteration [3], and cease the AL process once the model performance reaches 90% of the upper-bound performance. Experimental results are depicted in Figure 5.[4] As observed, XAL requires the least amount of data to reach the performance goal. For instance, in the DEBA task, XAL necessitates an average of 461.11 data points, which is 55.56 less than the second lowest value (BK–516.67). To conclude, XAL models only require 6%, 3%, 16%, and 10% of the data from RTE, MRPC, COVID19, and DEBA tasks respectively to achieve 90% performance of models

---

[3]To balance the training efficiency and the performance gap, we set the selection number as 50.

[4]For ease of presentation and training efficiency, we only report results on four tasks.

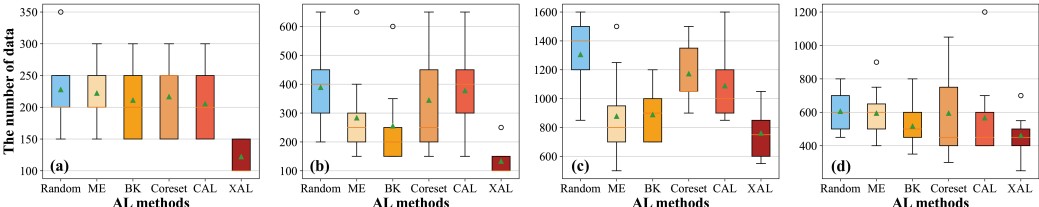

Figure 5: Experimental results demonstrate how much data, when selected using AL methods, is required for the models to achieve 90% of the performance of those trained on the complete training datasets. In each iteration, we annotate 50 instances. The performance of models trained on the whole training sets is, (a) RTE – 83.11%, (b) MRPC – 84.74%, (c) COVID19 – 75.45%, and (d) DEBA – 65.71%. The green triangles refer to the average values of the experiments on three different initial sets $D_l$ and three different random seeds. The circles refer to outliers. Detailed results can be seen in Appendix D.2.

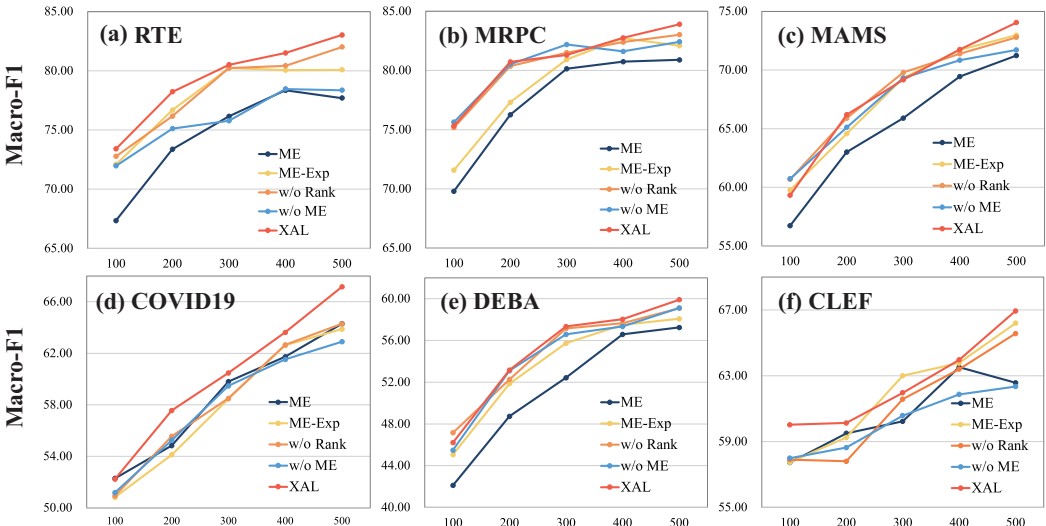

Figure 6: Results of ablation study in the six text classification tasks. We select 100 instances in each iteration and conduct 4 iterations (the same with Section 4.1). The results are measured using macro-F1 scores and they are the average values on three different initial sets $D_l$ and three different random seeds.

that are trained on the entire datasets, which significantly reduces the annotation cost. These results show that the proposed XAL is very cost-efficient in selecting informative unlabeled data.

## 4.2 ABLATION STUDY

We conduct an ablation study to investigate the impact of each module in our model. The results are displayed in Figure 6. Firstly, we conduct a comparison among ME, ME-Exp, and XAL, where ME-Exp has the same model structure as XAL but it selects the unlabeled data with the predicted classification entropy. We observe that ME-Exp can achieve superior performance on most datasets compared to ME, which demonstrates the effectiveness of using explanations. However, XAL further achieves noticeably better performance over ME-Exp, indicating that the improvement in XAL comes not only from the introduction of explanations but also from the data selection strategy (with explanation scores). Next, we compare XAL with a version that removes the ranking loss (*w/o Rank* in Figure 6). XAL also achieves better performance on most datasets and with different numbers of labeled data, indicating that the ranking loss can enhance the effectiveness of data selection in the AL process. Furthermore, the performance of selecting data solely using the explanation score but without using predictive entropy is also illustrated in Figure 6 (*w/o ME*). We observe that removing ME leads to significant performance drops on most datasets, implying that the predictive entropy and explanation score can complement each other.

To further evaluate how the ranking loss works in XAL, we also compare the model's capability to rank explanations between XAL and its counterpart without ranking loss. Experimental results show that XAL achieves superior performance. For instance, the ranking accuracy in RTE and MRPC for XAL are 73.93% and 78.62%, which are 5.36% and 4.30% higher than those without ranking loss, respectively (more details are shown in Appendix D.4). These results suggest that the ranking loss can enhance the model's ability to score the explanations.

## 4.3 EXPLANATION GENERATION

We also carry out experiments to analyze how the generation of explanations impacts model performance. Specifically, we replace ChatGPT with ALPACA-7B (Taori et al., 2023), and GPT4 [5] to generate explanations on the MAMS dataset. The results are presented in Table 2. It's noteworthy that the model using GPT4 generation achieves superior performance compared to the one using ChatGPT, suggesting that GPT4 can generate more useful

|  | 100 | 200 | 300 | 400 | 500 |
|---|---|---|---|---|---|
| ME | 52.75 | 57.45 | 63.72 | 66.28 | 69.15 |
| ChatGPT | 60.79 | 63.47 | 68.51 | 71.54 | 73.24 |
| ALPACA-7B | 59.52 | 61.75 | 67.77 | 71.12 | 72.24 |
| GPT4 | 59.67 | 64.28 | 69.51 | 72.96 | 74.63 |

Table 2: Model performance on MAMS using different explanation generations. We compare the performance in a certain initial set and random seed.

and informative explanations in XAL. We also observe that the ALPACA-7B can also provide useful explanations to some extent and enhance the model performance compared with ME. This suggests that LLMs, when used as an assistant in XAL, can provide consistent explanation generation and enhance model performance. The results of human annotation are also discussed in Appendix E.

## 4.4 HUMAN EVALUATION ON INTERPRETABILITY

We evaluate our model's ability to explain its prediction by examining the consistency between the generated explanation and the classification label. Specifically, we randomly select 50 test instances and use the model trained on 500 instances (see Section 4.1) to generate the labels and explanations. Then we ask humans to infer the classification labels based solely on the generated explanations. The consistency is measured by whether the human-inferred label equals the label predicted by the model. We report the consistency rate across all the test sets: MRPC-94%, RTE-94%, COVID19-96%, DEBA-94%, MAMS-94%, CLEF-100%. We find that the consistency rates on all six tasks exceed 94%, which demonstrates that XAL explains its classification prediction very well. Case studies for the generated explanations and the predicted labels are presented in Appendix F.

## 4.5 REPRESENTATION VISUALIZATION

To understand the potential of XAL in exploring informative unlabeled data, we use t-SNE (van der Maaten & Hinton, 2008) to "visualize" the data selection procedure of ME and XAL on the task DEBA. Specifically, with the intermediate model in Section 4.1 (trained with 200 labeled instances), 100 instances from the unlabeled set $D_u$ are then selected for annotation. Then, we feed all the labeled and unlabeled instances into the model and get their sentence representations ($\mathbf{h}_0$ in Eq. 1). Finally, we apply the t-SNE toolkit to map these representations into a two-dimensional embedding space, which is shown in Figure 8. We can observe that the unlabeled data selected by ME is only distributed around the decision boundary, which shows that the model can only select the high-uncertainty data it believes. However, the proposed XAL can select more diverse data, some of which are wrongly classified by the current model. These results demonstrate that the data selection strategy in XAL can identify more informative data. More visualizations are shown in Appendix G.

## 5 RELATED WORK

**Text Classification**. Recently, LLMs of generative schema have shown excellent performance in various NLP tasks including text classification (Min et al., 2022; Møller et al., 2023). However, some studies show that in-context learning based on LLMs (Radford et al., 2019; Brown et al., 2020)

---

[5] https://openai.com/gpt-4

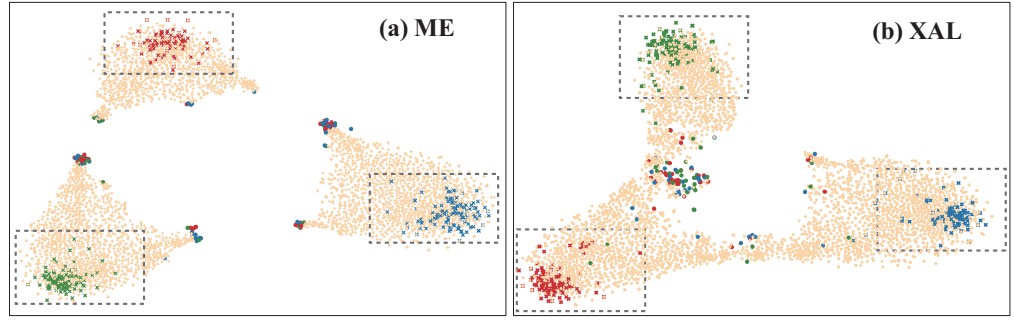

Figure 7: t-SNE visualizations of contextual representations in the data selection process on DEBA. To facilitate identification, we have outlined the areas of labeled data with dashed squares. The colors, i.e. red, blue, and green, refer to the golden labels of positive, negative, and neutral, respectively.

suffers from practical issues such as high computation costs for inference (Liu et al., 2022), over-sensitive to example choices and instruction wording (Gao et al., 2020; Schick & Schütze, 2021) uninterpretable behaviors (Zhao et al., 2021; Min et al., 2022). How to develop small and explainable classifiers is still in demand, since LLMs can also be easily confused and uninterpretable.

**Active Learning** is widely studied in the natural language processing area, ranging from text classification (Roy & McCallum, 2001; Zhang et al., 2017; Maekawa et al., 2022), and sequence labeling (Settles & Craven, 2008) to text generation (Zhao et al., 2020). We mainly focus on the task of text classification in this paper. Previous methods can be roughly divided into informativeness-based selection strategies, representativeness-based selection strategies, and hybrid selection strategies (Zhang et al., 2022a). The most mainstream methods, i.e., informativeness-based methods, are mostly characterized using model uncertainty, disagreement, or performance prediction, which suffers from over-confidence and a lack of exploration (Guo et al., 2017; Margatina et al., 2021). On the other hand, the representativeness-based methods rely on model inputs such as the representations of texts, which tends to select simple data samples and results in unsatisfactory performance (Roy & McCallum, 2001; Margatina et al., 2021).

**Explanation Information**, as external knowledge, has been proven useful for a wide range of tasks in natural language processing (Hase & Bansal, 2022). Hase et al. (2020) used explanations as additional information and directly fed them into models. Narang et al. (2020) and Shen et al. (2023) took the explanations as outputs and trained NLP models to generate them. How to leverage explanations is still an open problem (Hase & Bansal, 2022). In the active learning schema, some studies also attempt to leverage the explanations (Liang et al., 2020; Wang et al., 2021), but they mainly focus on promoting the generalization abilities of models trained on low-resource data. For example, Liang et al. (2020) uses external explanation to optimize the decision boundary between labels for image classification using a semantic knowledge grounding module. Unlike the aforementioned studies, in this paper, we explore how to leverage the explanations to identify informative unlabeled data for annotation.

## 6 CONCLUSION

In this paper, we proposed a novel Explainable Active Learning (XAL) framework for text classification, which aims to encourage classifiers to justify their inferences and delve into unlabeled data for which they cannot provide reasonable explanations. Experiments demonstrated that XAL achieves substantial improvements compared with previous AL methods. Despite achieving much improvement, we still notice that there are some limitations in our method (see Appendix H). In the future, we will try to handle these limitations and test our methods on more tasks.

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

# A    TASKS AND CORRESPONDING PROMPTS

We show the tasks and examples for experiments in Table 3, including natural language inference, paraphrase detection, category sentiment classification, stance detection, (dis)agreement detection, and relevance classification. We also show the prompts we used for explanation generation through querying ChatGPT (Table 4).

| Task | Text |
|------|------|
| Natural Language Inference | **Sentence 1**: Danny Kennedy, Greenpeace campaigns director, said: "The burden of proof in the Scott Parkin expulsion case lies morally with the Commonwealth, to prove that he is a danger. When the Government brought in anti-terror legislation, they promised the public that these laws would only be used to confront a real and present risk of a terrorist attack, not a sweep-all approach against citizens. Peace is not terrorism. Peace is not a threat to national security. No democratic government should expel a foreign citizen because [it] opposes his political opinions." **Sentence 2**: Greenpeace director said that peace is terrorism. **Label**: Not Entailment. |
| Paraphrase Detection | **Sentence 1**: Last week the power station's US owners, AES Corp, walked away from the plant after banks and bondholders refused to accept its financial restructuring offer .", **Sentence 2**: "The news comes after Drax's American owner, AES Corp. AES.N, last week walked away from the plant after banks and bondholders refused to accept its restructuring offer. **Label**: Paraphrase/Semantic Equivalent. |
| Category Sentiment Classification | **Text:** I left feeling unsatisfied, except for having a nice chance to people watch in the cozy atmosphere with my over-priced pasta bolognese. **Target**: Ambience **Label**: Positive |
| Stance Detection | **Text:** Michigan is fining individuals 500\$ for not wearing a mask in public. How do y'all feel about this? Curious because I am torn about being so forceful but agree that people should wear masks. #MaskOn. **Target**: Face Mask **Label**: Favor |
| (Dis)agreement Detection | **Text 1**: True, but with lower power usage, you have less heat to dissipate, meaning you can overclock it even more. **Text 2**: AMD creates a chip that saves energy by over 31 times. Someone show this to r/PCMasterRace cause we need to switch to AMD. **Label**: Agree. |
| Relevance Classification | **Document:** 99mtechnetium penicillamine: a renal cortical scanning agent. 99mTechnetium penicillamine, a renal cortical imaging agent, can be used to provide a rapid, safe, and non-invasive assessment of renal morphology and the renal vascular supply. Since this agent is not excreted significantly during the imaging procedure cortical scans of high quality can be obtained without image deterioration owing to a superimposed collecting system. These scans, which are clearly superior in anatomical detail to earlier scans using 131I hippuran, can be obtained along with the 131I hippuran renogram when the patient comes to the nuclear medicine department. Herein we demonstrate the anatomical detail it is now possible to achieve by presenting the cortical renal scans and accompanying radiograms from 5 patients with different renal pathology. **Topic**: Procalcitonin, C-reactive protein, and erythrocyte sedimentation rate for the diagnosis of acute pyelonephritis in children. **Label**: Not Relevant. |

Table 3: Tasks and examples for experiments.

| Task | Prompts | Label set |
|---|---|---|
| Natural Language Inference | Sentence 1: **{Text 1}**. Sentence 2: **{Text 2}**. Sentence 1 can **{Label}** sentence 2, explain the reason within 50 words. | {Entail, Not Entail} |
| Paraphrase Detection | Sentence 1: **{Text 1}**. Sentence 2: **{Text 2}**. The relation between the above two sentences is **{Label}**, explain the reason within 50 words. | {Paraphrase Equivalent, Not Paraphrase Equivalent} |
| Category Sentiment Classification | **{Text}**. | {Positive, Negative, Neutral} |
| Stance Detection | **{Text}**. The stance of this tweet to the target **{Target}** is **{Label}**, explain the reason within 50 words. | {Favor, Neutral, Against} |
| (Dis)agreement Detection | Sentence 1: **{Text 1}**. Sentence 2: **{Text 2}**. The stance of sentence 2 is **{Label}** to sentence 1, explain the reason within 50 words. | {Agree, Neutral, Disagree} |
| Relevance Classification | **{Document}**. The scientific document is **{Label}**' to the research topic **{Topic}**, explain the reason within 100 words. | {Relevant, Not Relevant} |

Table 4: The prompts we adopted for the tasks in our experiments.

# B EXPLANATION EXAMPLES

# C ALGORITHM

We show the detailed algorithm of XAL in Algorithm 1.

---

**Algorithm 1** Explainable Active Learning Algorithm

---

1: Initialization: dataset $D_u$, iteration steps $\mathcal{M}$, selective number $L$, training epoch $\mathcal{T}$.
2: Randomly select $L$ data from $D_u$, denoted as $D_s$ and remove them in $D_u$.
3: Annotate the data $\mathbf{x}_i \in D_s$ for $y_i^c$ with human annotators.
4: Query ChatGPT for diverse explanations $y_i^{g_r}$ for the data $\{\mathbf{x}_i, y_i^c\} \in D_s$.
5: Add $\{\mathbf{x}_i, y_i^c, y_i^{g_r}\} \in D_s$ to $D_l$, and empty the set $D_s$.
6: $m = 1$.
7: **repeat**
8:    $m \leftarrow m + 1$
9:    Initialize an explainable classifier $\pi$ and $t = 0$.
10:    **repeat**
11:       $t \leftarrow t + 1$
12:       Calculate optimization loss using data $\{\mathbf{x}_i, y_i^c, y_i^{g_r}\} \in D_l$.
13:       Optimize the explainable classifier $\pi$.
14:    **until** $t > \mathcal{T}$
15:    Calculate the predictive entropy $\mathbf{p}_i$ and explanation scores $\mathbf{c}_i$ of data $\mathbf{x}_i \in D_u$ uing Eq. 6.
16:    Calculate the rank score using Eq. 8.
17:    Select $L$ data with the largest score from $D_u$ to $D_s$.
18:    Annotate the data in $D_u$ following the steps 3-5.
19: **until** $m > \mathcal{M}$
**Output:** Explainable classifier $\pi$.

---

| Input | Label | Explanation |
|---|---|---|
| **RTE**
**Sentence 1**: The main institutionalised forms of recognition for those who have made a significant contribution in the fields of physics, chemistry, medicine, literature, as well as for those working for peace (and more recently in the area of economics), are the Nobel prizes.
**Sentence 2**: Nobel Peace Prize candidates have been chosen. | **Not Entailment** | Sentence 1 states that the Nobel prizes are the main forms of recognition for various fields, including peace. However, sentence 2 only mentions that candidates have been chosen for the Nobel Peace Prize, without specifying if they have actually received the prize. Therefore, sentence 2 does not necessarily follow from sentence 1. |
| | Entailment | Sentence 1 can entail sentence 2 because sentence 1 mentions that one of the main forms of recognition for those working for peace is the Nobel prizes. Therefore, it is reasonable to assume that candidates for the Nobel Peace Prize have been chosen. |
| **MRPC**
**Sentence 1**: Maddox, 87, cracked two ribs when he fell about 10 days ago at an assisted living home where he was recovering from intestinal surgery, Virginia Carnes said.
**Sentence 2**: Maddox, who had battled cancer since 1983, cracked two ribs earlier this month when he fell at an assisted living home where he was recovering from surgery. | **Equivalent** | The two sentences provide similar information but are expressed differently. Sentence 1 mentions that Maddox fell about 10 days ago, while Sentence 2 specifies that the fall occurred earlier this month. Despite the slight difference in wording, both sentences convey the same meaning. |
| | Not Equivalent | The reason why the two sentences are not paraphrases or semantic equivalents is because they provide different information about the timing and reason for Maddox's rib injury. In the first sentence, it mentions that Maddox fell about 10 days ago while recovering from intestinal surgery, whereas the second sentence states that he fell earlier this month while recovering from surgery without specifying the exact time frame. |
| **COVID19**
**Tweet:** Placed2nd @AngelaBelcamino Omg people say I have to wear a mask and I'm crying because I have to wear a mask to save lives boo hoo!! MAGAts are stupid!!! | **Favor** | The stance of this tweet towards face masks is favorable. The reason for this is that the tweet expresses support for wearing masks to save lives and criticizes those who oppose wearing masks, referring to them as ""MAGAts"" (a term often used to describe supporters of former President Donald Trump)." |
| | Against | The reason behind the stance against face masks in this tweet seems to be frustration and ridicule towards those who oppose wearing masks. The tweet suggests that the person finds the idea of wearing masks to save lives unnecessary and mocks those who support it, particularly referring to them as ""MAGAts,"" a derogatory term for supporters of former President Trump. |
| | Neutral | The tweet expresses a neutral stance towards face masks by mocking the idea of crying over having to wear one. While it criticizes MAGA supporters, it does not explicitly state a negative or positive opinion about face masks themselves." |

Table 5: Diverse explanations generated from ChatGPT. The golden label is marked in **bold**.

# D    Detailed Results

## D.1    Main Results

|  | Random | ME | BALD | BK | LC | Coreset | CAL | XAL |
|---|---|---|---|---|---|---|---|---|
| ***RTE*** | | | | | | | | |
| 100 | 67.34 | 67.34 | 67.34 | 67.34 | 67.34 | 67.34 | 67.34 | **73.40** |
| 200 | 70.64 | 73.37 | 68.06 | 72.80 | 72.91 | 71.12 | 72.22 | **78.22** |
| 300 | 72.16 | 76.15 | 71.09 | 76.85 | 75.60 | 74.90 | 74.81 | **80.51** |
| 400 | 74.71 | 78.35 | 73.11 | 76.90 | 78.15 | 75.16 | 76.15 | **81.50** |
| 500 | 75.54 | 77.69 | 74.30 | 79.44 | 79.22 | 77.69 | 75.42 | **83.02** |
| | | | | | | | | |
| ***MRPC*** | | | | | | | | |
| 100 | 69.80 | 69.80 | 69.80 | 69.80 | 69.80 | 69.80 | 69.80 | **75.31** |
| 200 | 75.44 | 76.26 | 73.95 | 76.35 | 77.10 | 76.54 | 76.22 | **80.73** |
| 300 | 78.12 | 80.14 | 76.07 | 80.23 | 79.87 | 79.39 | 78.52 | **81.31** |
| 400 | 80.28 | 80.74 | 77.64 | 81.95 | 81.21 | 79.85 | 79.76 | **82.76** |
| 500 | 80.63 | 80.90 | 79.90 | 82.33 | 81.53 | 80.06 | 80.60 | **83.91** |
| | | | | | | | | |
| ***MAMS*** | | | | | | | | |
| 100 | 56.73 | 56.73 | 56.73 | 56.73 | 56.73 | 56.73 | 56.73 | **59.32** |
| 200 | 61.77 | 63.01 | 58.75 | 62.34 | 62.83 | 62.59 | 61.89 | **66.19** |
| 300 | 66.38 | 65.90 | 62.68 | 66.92 | 66.72 | 64.83 | 65.96 | **69.16** |
| 400 | 67.88 | 69.44 | 64.33 | 69.67 | 69.74 | 68.93 | 67.54 | **71.74** |
| 500 | 70.05 | 71.23 | 66.69 | 71.78 | 71.83 | 69.50 | 69.59 | **74.04** |
| | | | | | | | | |
| ***COVID19*** | | | | | | | | |
| 100 | 52.29 | 52.29 | 52.29 | 52.29 | 52.29 | 52.29 | 52.29 | 52.24 |
| 200 | 57.19 | 54.84 | 53.19 | 57.22 | 55.67 | 56.18 | 55.67 | **57.57** |
| 300 | 57.95 | 59.80 | 54.74 | 60.10 | 58.45 | 58.18 | 58.45 | **60.48** |
| 400 | 59.85 | 61.73 | 55.98 | 62.30 | 61.38 | 60.62 | 61.38 | **63.63** |
| 500 | 61.78 | 64.30 | 56.01 | 64.36 | 64.48 | 61.45 | 64.48 | **67.16** |
| | | | | | | | | |
| ***DEBA*** | | | | | | | | |
| 100 | 42.09 | 42.09 | 42.09 | 42.09 | 42.09 | 42.09 | 42.09 | **46.21** |
| 200 | 50.60 | 48.74 | 46.81 | 50.65 | 49.73 | 49.18 | 49.26 | **53.16** |
| 300 | 53.93 | 52.43 | 51.54 | 54.87 | 54.57 | 53.97 | 53.43 | **57.35** |
| 400 | 57.03 | 56.58 | 53.18 | 57.02 | 57.15 | 56.37 | 55.06 | **58.03** |
| 500 | 57.45 | 57.25 | 55.66 | 57.78 | 57.64 | 56.95 | 55.82 | **59.91** |
| | | | | | | | | |
| ***CLEF*** | | | | | | | | |
| 100 | 57.72 | 57.72 | 57.72 | 57.72 | 57.72 | 57.72 | 57.72 | **60.02** |
| 200 | 57.95 | 59.50 | 57.38 | 59.82 | 58.44 | 59.62 | **60.67** | 60.13 |
| 300 | 57.37 | 60.23 | 58.80 | 60.66 | 61.72 | 59.60 | 58.44 | **61.97** |
| 400 | 60.04 | 63.52 | 59.14 | 63.48 | 63.81 | 61.46 | 59.04 | **63.97** |
| 500 | 61.04 | 62.57 | 59.03 | 64.66 | 65.16 | 59.84 | 61.53 | **66.94** |

Table 6: Main results in the six text classification tasks. We select 100 instances in each iteration and conduct 4 iterations. The results are measured using macro-F1 scores and they are the average values on three different initial sets $D_l$ and three different random seeds.

## D.2    Given upper bound

We show the average number of data required for the model to achieve 90% performance of those trained on all the training data.

## D.3    Ablation Study

## D.4    Capacity of Score

To assess our model's capability to distinguish between reasonable and unreasonable explanations, we evaluate its ranking performance on the test set. Specifically, after four iterations of the AL

|  | Random | ME | BK | Coreset | CAL | XAL |
|---|---|---|---|---|---|---|
| RTE | 388.89 | 283.33 | 255.56 | 344.44 | 377.78 | **133.33** |
| MRPC | 227.78 | 222.22 | 211.11 | 216.67 | 205.56 | **122.22** |
| COVID19 | 1305.56 | 877.78 | 888.89 | 1172.22 | 1088.89 | **761.11** |
| DEBA | 605.56 | 594.44 | 516.67 | 594.44 | 566.67 | **461.11** |

Table 7: The detailed experimental results about how much data queried by AL methods can the model achieve 90% performance of the models trained on the whole training data. In each iteration, we select 50 data. The model performances trained on the whole training sets are, (a) RTE – 83.11%, (b) MRPC – 84.74%, (c) COVID19 – 75.45%, and (d) DEBA – 65.71%. The green triangles refer to the average values of the nine-times experiments.

|  | ME | ME-Exp | w/o Rank | w/o ME | XAL |
|---|---|---|---|---|---|
| ***RTE*** |  |  |  |  |  |
| 100 | 67.34 | 72.09 | 72.77 | 71.96 | **73.40** |
| 200 | 73.37 | 76.68 | 76.17 | 75.10 | **78.22** |
| 300 | 76.15 | 80.23 | 80.22 | 75.77 | **80.51** |
| 400 | 78.35 | 80.05 | 80.42 | 78.46 | **81.50** |
| 500 | 77.69 | 80.08 | 82.01 | 78.36 | **83.02** |
| ***MRPC*** |  |  |  |  |  |
| 100 | 69.80 | 71.58 | 75.18 | 75.64 | **75.31** |
| 200 | 76.26 | 77.32 | 80.37 | 80.53 | **80.73** |
| 300 | 80.14 | 80.93 | 81.50 | **82.19** | 81.31 |
| 400 | 80.74 | 82.72 | 82.40 | 81.61 | **82.76** |
| 500 | 80.90 | 82.09 | 83.02 | 82.42 | **83.91** |
| ***MAMS*** |  |  |  |  |  |
| 100 | 56.73 | 59.77 | **60.69** | 60.73 | 59.32 |
| 200 | 63.01 | 64.57 | 65.90 | 65.11 | **66.19** |
| 300 | 65.90 | 69.32 | **69.79** | 69.32 | 69.16 |
| 400 | 69.44 | 71.71 | 71.38 | 70.83 | **71.74** |
| 500 | 71.23 | 72.97 | 72.79 | 71.71 | **74.04** |
| ***COVID19*** |  |  |  |  |  |
| 100 | **52.29** | 50.83 | 50.94 | 51.19 | 52.24 |
| 200 | 54.84 | 54.13 | 55.56 | 55.25 | **57.57** |
| 300 | 59.80 | 58.48 | 58.51 | 59.48 | **60.48** |
| 400 | 61.73 | 62.63 | 62.66 | 61.53 | **63.63** |
| 500 | 64.30 | 63.88 | 64.29 | 62.90 | **67.16** |
| ***DEBA*** |  |  |  |  |  |
| 100 | 42.11 | 45.06 | **47.16** | 45.48 | 46.21 |
| 200 | 48.74 | 51.86 | 52.26 | 53.11 | **53.16** |
| 300 | 52.43 | 55.74 | 57.15 | 56.59 | **57.35** |
| 400 | 56.58 | 57.51 | 57.65 | 57.35 | **58.03** |
| 500 | 57.25 | 58.08 | 59.09 | 59.11 | **59.91** |
| ***CLEF*** |  |  |  |  |  |
| 100 | 57.72 | 57.74 | 57.91 | 57.99 | **60.02** |
| 200 | 59.50 | 59.24 | 57.80 | 58.63 | **60.13** |
| 300 | 60.23 | 63.00 | 61.58 | 60.58 | **61.97** |
| 400 | 63.52 | 63.78 | 63.40 | 61.87 | **63.97** |
| 500 | 62.57 | 66.20 | 65.57 | 62.35 | **66.94** |

Table 8: Detailed results of ablation study in the six text classification tasks. We select 100 instances in each iteration and conduct 4 iterations. The results are measured using macro-F1 scores and they are the average values on three different initial sets $D_l$ and three different random seeds.

process as per section 4.1, we prompt ChatGPT to generate diverse explanations for the test data and score them using Eq. 6. In each test step, we feed both a reasonable and an unreasonable explanation to our model and calculate the accuracy in predicting the reasonable ones based on the computed

|  | RTE | MRPC | MAMS | COVID19 | DEBA | CLEF |
|---|---|---|---|---|---|---|
| w Ranking Loss | 73.93 | 78.62 | 68.64 | 62.25 | 59.78 | 90.73 |
| w/o Ranking Loss | 68.57 | 74.32 | 62.04 | 58.34 | 55.04 | 87.39 |

Table 9: The performance of scoring the explanations w/o ranking loss.

explanation score (Table 9). As seen in the results, the model incorporating ranking loss achieves superior performance compared to the model without it. For instance, the accuracy in RTE and MRPC are 73.93% and 78.62% in the model with ranking loss, which are 5.36% and 4.30% higher than those without ranking loss, respectively. The improvement in prediction accuracy suggests that the ranking loss can enhance the model's ability to score the reasonability of explanations.

## E   HUMAN ANNOTATION

We also carry out experiments to analyze how the human generation of explanations impacts model performance. Specifically, we replace ChatGPT with human annotation to generate explanations on the MAMS dataset. For human annotation, three PhD students specializing in NLP annotate the labels and explanations. Specifically, the models achieve the macro-F1 scores of RTE-62.13, MRPC-63.36, MAMS-67.38, COVID19-69.70, and CLEF-71.56, which are relatively lower compared to ChatGPT, which could be due to inconsistent annotation styles among annotators and changes in the annotation scheme from the original dataset (Gilardi et al., 2023; Zhu et al., 2023). The results also demonstrate the effectiveness of explanation generation through LLMs in XAL.

# F    CASE STUDY

## F.1    MODEL GENERATION

Some generation cases are shown from the models trained for 500 data in the AL process of Section 4.1. In these cases, we can find that our model can generate reasonable explanations for the label, which indicates the interpretability of our inference. But in some cases such as the case iv., although the explanation generates the correct label Agree, it explains the label with a wrong reason, which implies that the explainer does not perform perfectly in the small number of data. But it also indicates that we can enhance the model performance in inference and generation by selecting the data with unreasonable explanations through human beings.

| Input | Golden | Explanation |
| --- | --- | --- |
| **i. *RTE*** 
 **Sentence 1**: Dana Reeve, the widow of the actor Christopher Reeve, has died of lung cancer at age 44, according to the Christopher Reeve Foundation. 
 **Sentence 2**: Christopher Reeve had an accident. | Not Entailment | Sentence 1 states that Dana Reeve, the widow of Christopher Reeve, has died of lung cancer at the age of 44. It does not provide any information about what caused Christopher Reeve's accident. Therefore, sentence 1 does not entail sentence 2. |
| **ii. *MRPC*** 
 **Sentence 1**: Air Commodore Quaife said the Hornets remained on three-minute alert throughout the operation. 
 **Sentence 2**: Air Commodore John Quaife said the security operation was unprecedented. | Not Equivalent | The reason why the two sentences are not paraphrases or semantic equivalents is because they provide different information. Sentence 1 states that the Hornets were on three-minute alert during the operation, while Sentence 2 states that the security operation was unprecedented. |
| **iii. *MAMS*** 
 **Review**: Waiter took our drink order and then we didn't see him for 15 minutes. 
 **Category**: Stuff. | Negative | The reason for the negative sentiment towards the staff in this review is because the waiter took the drink order and disappeared after 15 minutes. This indicates a lack of professionalism and attentiveness, which can be frustrating for customers. |
| **iv. *COVID19*** 
 **Tweet**: @WCheesemaker @StephenKing No. You're missing the point. And from the way I phrased my statement I can understand why. I was attempting to point out masks work! They help stop the spread of disease. I should have stated it more clearly. I'm a scientist; a big fan of masks- and science!! :relaxed: 
 **Target**: Face mask. | Favor | The stance of this tweet is in favor of face masks. The reason given is that the tweeter believes masks are effective in stopping the spread of disease, and therefore should be encouraged to use them. |
| **v. *DEBA*** 
 **Sentence 1**: Best thing you can do is simply not drive. Take public transportation instead. 
 **Sentence 2**: That's hugely unrealistic for many people living in places with poor public transport provisions (rural areas, poorer areas) or who live in poorly planned towns where amenities are spatially distant. | Disagree | Sentence 2 disagrees with Sentence 1 because it argues that taking public transportation instead of driving is not feasible for many people living in places with poor public transport provisions or in poorly planned towns. |
| **vi. *DEBA*** 
 **Sentence 1**: I can't imagine believing Joe "Nothing Will Fundamentally Change" Biden is going to do anything except continue to serve his corporate masters. This is... uh, not a guy who's gonna accomplish much in office. 
 **Sentence 2**: Biden has surprised me with some of his action but on the whole it's still up to activists and local politicians to get stuff done. Shame | Agree | Sentence 2 agrees with Sentence 1 because it acknowledges that Biden has made some progress, but it also emphasizes that it is up to activists and local politicians to bring about meaningful change. |

Table 10: Case study of the explanation generation of our model. The model is trained on 500 labeled data instances following the AL process in Section 4.1.

## F.2 Unreasonable Generation

We also show some cases that our model believes have high unreasonability in the training set (Table 11). It is noted that in these cases the model generates some unreasonable explanations.

| Input | Golden | Explanation |
|---|---|---|
| **i. MAMS**
**Review**: I like the smaller portion size for dinner.
**Category**: miscellaneous | Positive | The reason for the neutral sentiment polarity towards the aspect "miscellaneous" in the review is that the reviewer prefers the smaller portion size for dinner. This suggests that the overall dining experience is more enjoyable and satisfying, resulting in a neutral sentiment polarity. |
| **ii. MAMS**
**Review**: After ordering drinks, we both decided on the Paella Vallenciana, brought out on hot plates.
**Category**: food | Neutral | The positive sentiment polarity of the review towards the aspect of food is due to the description of the Paella Vallenciana being brought out on hot plates. This suggests that the dish was served fresh and hot, enhancing the dining experience and leaving a positive impression. |
| **iii. COVID19**
**Tweet**: @Timewalkproject @BBCNormanS @BBCNews A lot less palaver than being hooked up to a life support machine! #WearA-Mask | Favor | The tweet expresses a negative stance towards face masks, referring to them as COVID-19. The reason for this stance is not provided within the given word limit. |
| **iv. DEBA**
**Sentence 1**: Energy mostly controls nuclear weapons and nuclear power plants, not fossil fuels. That's why they put a nuclear physicist (Muniz) in charge of it. What do people think Muniz going to do that will be harmful to the environment? Is this just an indirect anti-nuclear attack? Or just ignorance by insignificant, fringe environmental groups?
**Sentence 2**: DoE also has the National Renewable Energy Lab, where a lot of the tech development used for renewable energy has happened. The DoE funded the development of fracking. There's a history of recent appointees using positions there to champion the fossil fuels industry, even if most of what the DoE does is nuclear. | Agree | Sentence 2 disagrees with sentence 1 because it suggests that governments have not used taxes for the intended purposes. This implies that they raid the education budget to raise taxes for education, contradicting the claim in sentence 1 that the purpose of a carbon tax is to. |
| **v. DEBA**
**Sentence 1**: I can't imagine believing Joe "Nothing Will Fundamentally Change" Biden is going to do anything except continue to serve his corporate masters. This is... uh, not a guy who's gonna accomplish much in office
**Sentence 2**: And that will please you, because you'd rather complain about it and have your cynicism be validated. | Disagree | Sentence 2 agrees with Sentence 1 because it argues that the person who posts here is not trying to debate, but rather to annoy. It suggests that the solution is to simply out-annoy them and ban them. |

Table 11: Case study of the explanation generation of our model. The model is trained on 500 labeled data instances following the AL process in Section 4.1.

## G    REPRESENTATION VISUALIZATION

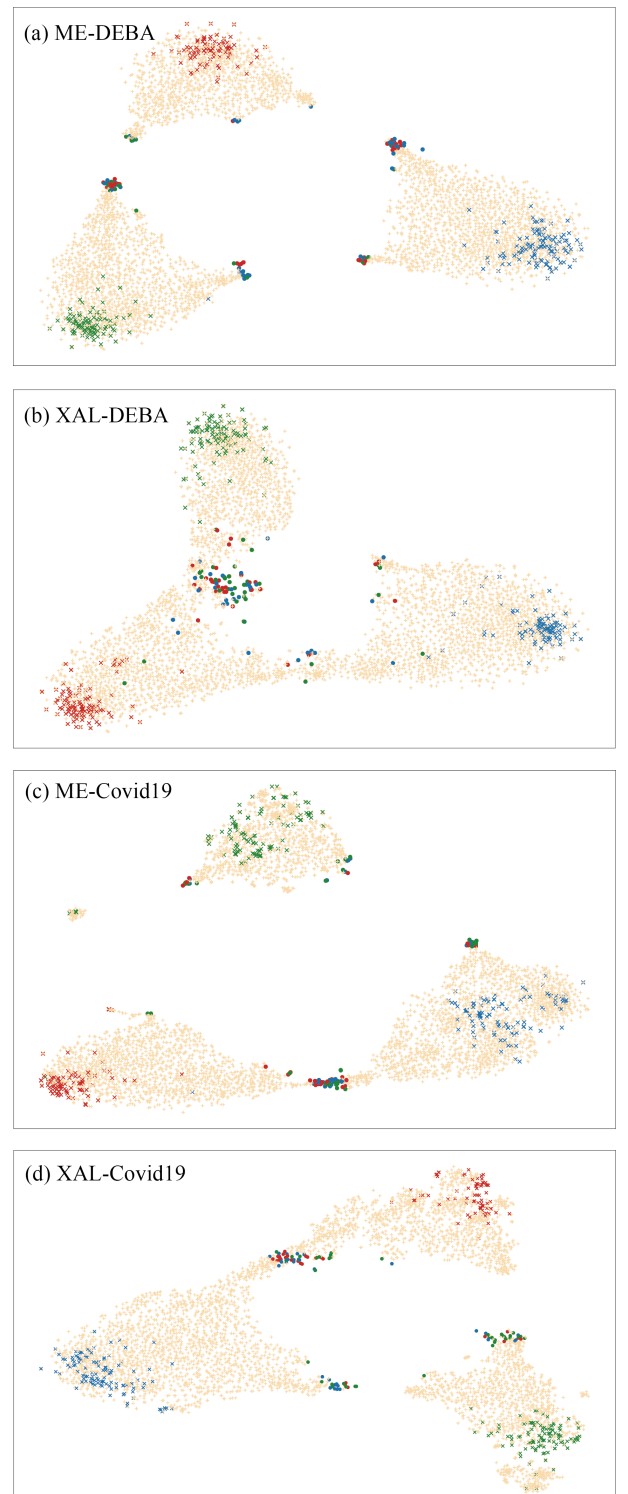

Figure 8: The t-SNE visualization of sentence representations in the data selection process.

## H    LIMITATIONS

Our model, which incorporates a decoder module in contrast to encoder-only classifiers, necessitates more time and computational resources for training. However, if the generation of an explanation is not required, we only need the encoder module for classification inference. In our experiments, we evaluated our model's effectiveness across six classification tasks in a low-resource setting, but XAL can be used for other tasks with more label classes and industrial downstream applications. It's important to note that the performance of XAL is somewhat dependent on the quality of the explanation (as demonstrated in Section 4.3). However, generating high-quality explanations necessitates access to more resource-intensive large language models. How to reduce the number of calls to LLMs is still an open and interesting research direction

