# OpenReview forum: "XAL: EXplainable Active Learning Makes Classifiers Better Low-resource Learners"
_ICLR.cc/2024/Conference — ICLR 2024 Conference Withdrawn Submission_

### Official Review · Reviewer_LLhf · 2023-10-31

**Soundness:** 2 fair
**Presentation:** 2 fair
**Contribution:** 3 good
**Rating:** 3
**Confidence:** 4

**Summary:**

This paper introduces an Explainable Active Learning framework tailored for text classification tasks in settings with limited resources. The key idea is to prompt the model to provide a natural language explanation for its prediction. This explanation is then leveraged both in active learning data selection and model fine-tuning.

**Strengths:**

The proposed method innovatively combines human-in-the-loop label annotation and LLM-in-the-loop explanation generation to optimize both the model performance and the annotation budget of active learning.

**Weaknesses:**

Section 3.1 It's inappropriate to use a development set for hyperparameter tuning, as highlighted in references [1][2]. This compromises the integrity of the experimental setup, and is the major reason for rejection.

Section 4.4: The human evaluation results are perplexing. The main results in Section 4.1 show the model's accuracy on all datasets to be considerably lower than 94%. Given that the predicted labels exhibit an accuracy range of 60-80%, a 94% consistency between the predicted label and its associated explanation appears contradictory.


paper
1. Weaker Than You Think: A Critical Look at Weakly Supervised Learning
2. On the Limitations of Simulating Active Learning

**Questions:**

Suggestions:
* In Section 2.4, change "golden" to "gold."
* In Figure 2, \pi is introduced without prior definition. Its definition is provided subsequently in Section 2.3.
* In Section 4.1, it would be beneficial to include the zero-shot ChatGPT results and the results from the model "trained on the entire training set" as flat lines in the figures.
* For Figure 2, ensure consistent color coding between the caption and the figure elements (e.g., red and blue arrows).

Questions:
* How about the performance of few-shot LLMs using a random select strategy, in addition to the zero-shot LLM?
* For Figure 4, all other baseline models demonstrate very similar performance across all datasets when limited to 100 data points. Could there be a specific reason underlying this?

---

> ### Author Response · Authors · 2023-11-21
> **Response to the Weakness**
>
> We greatly appreciate the careful comments and suggestions provided by the reviewer. Our response to the weakness is shown as follows:
>
> **Question 1**: Section 3.1 It's inappropriate to use a development set for hyperparameter tuning, as highlighted in references [1][2]. This compromises the integrity of the experimental setup, and is the major reason for rejection.
>
>  **Answer**: In alignment with the approach outlined in the referenced paper [1], we also utilize the development set solely for the purpose of selecting suitable checkpoints. It is a common strategy in the active learning area to use development set for selection of checkpoints [2-5]. Our set of hyperparameters is simply determined based on their magnitudes, with specific considerations such as the generative loss being at a value ten times higher than the classification loss.
>
> Furthermore, we adhere to a uniformity in hyperparameter settings across various tasks to mitigate the impact of differing parameter values. We will further incorporate a sensitivity analysis in the manuscript.
>
> **Question 2**: Section 4.4: The human evaluation results are perplexing. The main results in Section 4.1 show the model's accuracy on all datasets to be considerably lower than 94%. Given that the predicted labels exhibit an accuracy range of 60-80%, a 94% consistency between the predicted label and its associated explanation appears contradictory.
>
> **Answer**: The consistency rate metric assesses the alignment between the explanations provided by models and their corresponding classification labels, focusing on the support offered by explanations rather than the accuracy of the model inferences. This experiment is designed to showcase the model's proficiency in elucidating its predictions and highlighting its capability to provide coherent and supportive explanations for the assigned classification labels.
>
> **Question 3**: How about the performance of few-shot LLMs using a random select strategy, in addition to the zero-shot LLM?
>
> **Answer**: Few-shot Language Model (LLM) techniques have demonstrated the capacity to improve model performance to a certain extent. For example, in the task of Covid19, we implemented 5-shot in-context learning alongside queries to deduce stance labels. The results yielded a macro-F1 score of 67.73%, 1.06 higher than that of zero-shot. Meanwhile, In-Context Learning (ICL) does encounter challenges such as instability and uninterpretable behaviors, which indicates the importance of fine-tuning small models with effective use of labels. We will incorporate a detailed results of LLM few-shot performance in the manuscript.
>
> **Question 4**: For Figure 4, all other baseline models demonstrate very similar performance across all datasets when limited to 100 data points. Could there be a specific reason underlying this?
>
> **Answer**: The experiments maintained consistency by utilizing identical random seeds and initial labeled sets across all models. In contrast to other baseline models that solely leverage 100 data points along with their corresponding classification labels to fine-tune a classifier, our model adopts a distinctive approach. From the outset, our model takes into account both the classification labels and the generation of explanations. This unique combination results in a performance differential compared to other baselines, showcasing the added value of incorporating explanation generation from the initial stages of the model's training process.
>
> [1] Margatina K, Aletras N. On the Limitations of Simulating Active Learning[J]. arXiv preprint arXiv:2305.13342, 2023.
>
> [2] Yuan M, Lin H T, Boyd-Graber J. Cold-start Active Learning through Self-supervised Language Modeling[C]//Proceedings of the 2020 Conference on Empirical Methods in Natural Language Processing (EMNLP). 2020: 7935-7948.
>
> [3] Shujian Zhang, Chengyue Gong, Xingchao Liu, Pengcheng He, Weizhu Chen, and Mingyuan Zhou. 2022. ALLSH: Active Learning Guided by Local Sensitivity and Hardness. In Findings of the Association for Computational Linguistics: NAACL 2022, pages 1328–1342, Seattle, United States. Association for Computational Linguistics.
>
> [4] Margatina K, Vernikos G, Barrault L, et al. Active Learning by Acquiring Contrastive Examples[C]//Proceedings of the 2021 Conference on Empirical Methods in Natural Language Processing. 2021: 650-663.
>
> [5] Sener O, Savarese S. Active Learning for Convolutional Neural Networks: A Core-Set Approach[C]//International Conference on Learning Representations. 2018.

---

### Official Review · Reviewer_XVNV · 2023-10-31

**Soundness:** 3 good
**Presentation:** 3 good
**Contribution:** 2 fair
**Rating:** 5
**Confidence:** 4

**Summary:**

The manuscript proposes an Explainable Active Learning (XAL) framework for text classification with a pre-trained uni-directional decoder to generate and score the explanations. XAL proposes a ranking loss to enhance the decoder's capability in scoring explanations and combines the predictive uncertainty of the encoder and the explanation score of the decoder to select the most informative data for annotation. XAL is evaluated on 6 text classification datasets and the results show that the proposed method outperforms existing AL techniques.

**Strengths:**

XAL generates high-quality explanations for its classification decisions, which can better help users understand and trust the model. This combination makes Active Learning, as human-in-the-loop learning becomes more realistic.

**Weaknesses:**

- XAL only compares with typical Active Learning methods and does not compare with similar methods like [r1].

- As shown in the ablation study, each component could not provide a stable performance gain on various tasks, e.g., XAL vs. w/o rank.


[r1] Ghai B, Liao Q V, Zhang Y, et al. Explainable active learning (xal) toward ai explanations as interfaces for machine teachers[J]. Proceedings of the ACM on Human-Computer Interaction, 2021, 4(CSCW3): 1-28.

**Questions:**

- How to set the hyperparameters? Like $\lambda$, $\lambda_1$, $\lambda_2$.

- In experimental settings, the author only uses 500 labeled samples, as shown in Figure 4, most model performances are far from convergence.

- In figure 4, why the starting point (initial model performance) of XAL are different from other baselines?

---

> ### Author Response · Authors · 2023-11-22
> **Response to the Weakness**
>
> We greatly appreciate the careful comments and suggestions provided by the reviewer. Our response to the weakness is shown as follows:
>
> **Quesion 1**: XAL only compares with typical Active Learning methods and does not compare with similar methods like [r1].
>
> **Answer**: Indeed, integrating explanations into text classification tasks poses unique challenges compared to feature-based tasks. The method presented in the referenced paper [1], which predicts labels based on features such as age, gender, and education, relies on local feature importance calculated from the coefficients of a logistic regression model. However, applying this approach directly to text classification is challenging due to the inherently different nature of textual data. Text classification often involves complex and high-dimensional data, making it less amenable to traditional feature-based explanation methods. The nuances of language and the intricate relationships between words can make it difficult to attribute model decisions solely to individual features. In light of these challenges, our approach focuses on leveraging explanations specifically tailored for text classification tasks. By considering the unique characteristics of textual data, our model aims to provide interpretable and meaningful explanations that capture the linguistic nuances inherent in natural language processing tasks.
>
> **Question 2**: As shown in the ablation study, each component could not provide a stable performance gain on various tasks, e.g., XAL vs. w/o rank.
>
> **Answer**: The comprehensive results, available in Appendix D.3, reveal the noteworthy performance of our model in comparison to other methods, particularly ME-Exp and w/o rank, across various tasks. It is evident that our model consistently outperforms these alternatives in most time. However, it's crucial to recognize that the impact of different components on the final performance may vary for each task, resulting in some fluctuations in performance across different scenarios.
>
> **Question 3**: How to set the hyperparameters?
>
> **Answer**: We adhere to a methodology where the development set is not specifically employed for hyperparameter tuning. The selection of hyperparameters is guided by considerations of different orders of magnitude. For instance, we set the generative loss to be 10 times higher than the classification loss. Furthermore, we maintain consistency in hyperparameter configurations across different tasks to mitigate the potential impact of varying values. We will add sensitivity analysis in the manuscripts to delve deeper into the effects of hyperparameter variations.
>
> **Question 4**: In figure 4, why the starting point (initial model performance) of XAL are different from other baselines?
>
> **Answer**: The experiments maintained consistency by utilizing identical random seeds and initial labeled sets across all models. In contrast to other baseline models that solely leverage 100 data points along with their corresponding classification labels to fine-tune a classifier, our model adopts a distinctive approach. From the outset, our model takes into account both the classification labels and the generation of explanations. This unique combination results in a performance differential compared to other baselines, showcasing the added value of incorporating explanation generation from the initial stages of the model's training process.
>
> [1] Ghai B, Liao Q V, Zhang Y, et al. Explainable active learning (xal) toward ai explanations as interfaces for machine teachers[J]. Proceedings of the ACM on Human-Computer Interaction, 2021, 4(CSCW3): 1-28.

---

### Official Review · Reviewer_idfx · 2023-11-05

**Soundness:** 2 fair
**Presentation:** 2 fair
**Contribution:** 2 fair
**Rating:** 5
**Confidence:** 4

**Summary:**

Active Learning represents a potent learning paradigm that seeks to minimize the labelled data necessary for training while maximizing the model's performance. Central to active learning research is the design of an effective acquisition function, aimed at selecting the most informative samples with respect to the decision boundaries, thereby enhancing the classifier's learning process. This paper suggests leveraging the predictive explanations generated by Large Language Models (LLMs) such as ChatGPT to complement the predictive uncertainty computed through entropy. The acquisition function combines the weighted sum of entropy and the likelihood of the most probable explanation. In addition to the traditional cross-entropy loss, the authors introduce two additional components: an explanation generation loss and a ranking loss. The experimental results demonstrate the effectiveness of the proposed framework, showcasing its potential to deliver promising results.

**Strengths:**

* Originality:  The incorporation of explanations generated by LLMs for guiding the selection of informative samples is indeed an interesting idea, even though it is not completely new. This approach is thoughtfully motivated by the principles of explanation-based teaching and learning. Leveraging textual explanations from well-trained and established LLMs, such as ChatGPT, has the potential to provide valuable additional insights that can enhance the classifier's capability to distinguish various class patterns. The experimental results further validate its effectiveness in selecting samples for labelling.
* Clarity: In general, the paper effectively communicates its primary idea, albeit with some minor errors and issues that I will elaborate on below.
* Significance: The presented active learning framework yields promising results when compared to various baselines, such as BALD, CAL, LC, and others. Additionally, the ablation studies effectively demonstrate the contributions of different components within the proposed framework. The qualitative analysis employing T-SNE provides some valuable insights.

**Weaknesses:**

Despite the comprehensive demonstration of the proposed framework's performance across six different text classification tasks, it is noteworthy that the chosen baseline methods appear to be somewhat outdated. Recent developments in the field of active learning, including ALPS[2], BADGE [1], WMOCU [3], SoftMoCU [4], and BEMPS [5], could have provided more up-to-date benchmarks. Furthermore, the omission of works with similar explanation-based learning concepts mentioned in the related work section raises questions about the completeness of the experimental evaluation. Taken together, these factors lead to concerns regarding the sufficiency of the experimental work in demonstrating the advantages of the proposed framework.

Eq (7) has two running parameters and Eq(8) has one.The authors stated that those parameters were chosen empirically based on the preliminary experiments.  However, the reviewer thinks empirically running those running parameters has an implication in its adaptation, as how sensitive of the performance of the proposed active learning framework is unknown. Thus, it would be good to study the impact of those running parameters.

Furthermore, the ablation studies show that ME-Exp and w/o rank compare favourably with each other, even with XAL in some data sets. Considering the differences among the three models/variants, the review thought the second loss term in Eq(7) associated with the explanation generation might contribute substation ally to the ultimate performance difference. Adding the ablation studies to that term becomes essential together with the running parameters above. Meanwhile, there is a lack of studies on the acquisition batch size.

Interestingly, the authors have not conducted a comparative analysis of the computational costs associated with evaluating the acquisition functions of different active learning schemes. It would be of practical significance to assess the computational expenses involved in these methods.


References
* [1] J. T. Ash, C. Zhang, A. Krishnamurthy, J. Langford, and A. Agar- wal, “Deep batch active learning by diverse, uncertain gradient lower bounds,” in Proc. 8th Int. Conf. Learn. Representations, 2020.
* [2] 	M. Yuan, H.-T. Lin, and J. Boyd-Graber, “Cold-start active learning through self-supervised language modeling,” in Proc. 2020 Conf. Empirical Methods Natural Lang. Process. (EMNLP), Nov. 2020, pp. 7935– 7948.
* [3] G. Zhao, E. Dougherty, B.-J. Yoon, F. Alexander, and X. Qian, “Uncertainty-aware active learning for optimal Bayesian classifier,” in Proc. 9th Int. Conf. Learn. Representations, 2021.
* [4] G. Zhao, E. Dougherty, B.-J. Yoon, F. J. Alexander, and X. Qian, “Bayesian active learning by soft mean objective cost of uncertainty,” in Proc. 24th Int. Conf. Artif. Intell. Statist., vol. 130, Apr. 2021, pp. 3970–3978.
* [5] W.Tan, L.Du, and W.Buntine,“Diversityenhancedactivelearningwith strictly proper scoring rules,” in Advances in Neural Information Processing Systems, 2021, pp.10906– 10918.
* [6] Kuhn, L., Gal, Y. and Farquhar, S., 2022, September. Semantic Uncertainty: Linguistic Invariances for Uncertainty Estimation in Natural Language Generation. In The Eleventh International Conference on Learning Representations.

**Questions:**

* There are multiple factors that contribute to the performance of the proposed XAL method. The review wondered if the authors could show the convergence analysis of the active learner. In other words, can the learner guarantee to converge to the optimal classifier, as the number of acquired samples goes to infinity?
* The experimental results show that using the explanation score in acquiring samples can contribute to learning. What type of uncertainty does the proposed generation score capture? Is it something to do with semantic uncertainty[6]?
* Regarding Figure 3, Should B in the right column correspond to C in the left column?
* In Section 3.3, Should $D_u$ be $D_l$

**Details Of Ethics Concerns:**

Pushing data to service, like ChatGPT, always raises concerns about data privacy. The datasets used in the paper might all be publically available datasets. However, if one would like to use the method to privacy-sensitive domains, like medical corpora, the method proposed thus cannot be used.

---

> ### Author Response · Authors · 2023-11-22
> **Response to the Weakness**
>
> We greatly appreciate the careful comments and suggestions provided by the reviewer. Our response to the weakness is shown as follows:
>
> **Question 1**: It is noteworthy that the chosen baseline methods appear to be somewhat outdated. Recent developments in the field of active learning, including ALPS[2], BADGE [1], WMOCU [3], SoftMoCU [4], and BEMPS [5], could have provided more up-to-date benchmarks. Furthermore, the omission of works with similar explanation-based learning concepts mentioned in the related work section raises questions about the completeness of the experimental evaluation.
>
> **Answer:** We will add the baselines to our paper. For example, we have carried out experiments using BADGE[1], ALPS[2],  and BEMPS (CoreMSE) [3] on the task of MAMS. The results are shown as follows:
>
> |  | ALPS | BADGE | BEMPS | XAL |
> | --- | --- | --- | --- | --- |
> | 100 | 57.29 | 57.29 | 57.29 | 59.32 |
> | 200 | 60.44 | 60.91 | 62.04 | 66.19 |
> | 300 | 61.57 | 62.14 | 64.83 | 69.16 |
> | 400 | 64.69 | 67.13 | 68.61 | 71.74 |
> | 500 | 67.83 | 69.08 | 71.22 | 74.04 |
>
> As we observe, our model can still achieve a more significant performance. But the explanation-based learning methods mentioned in our related work are hard to implemented in the text classification task. For example, Alice [1] incorporates b explanations into a b class-image classification task, and it grounds the extracted knowledge on the training data by cropping the corresponding semantic segments, which is hard to implemented since there is not image patches for text classification. We are the first to consider explanations in the active learning scheme for text classification tasks.
>
> **Question 2:** However, the reviewer thinks empirically running those running parameters has an implication in its adaptation, as how sensitive of the performance of the proposed active learning framework is unknown. Thus, it would be good to study the impact of those running parameters.
>
> **Answer:** The selection of hyperparameters is guided by considerations of different orders of magnitude. For instance, the generative loss is 10 times higher than the classification loss.  We do not specially select the hyperparameter through grid search. Furthermore, we maintain consistency in hyperparameter configurations across different tasks to mitigate the potential impact of varying values. We will add sensitivity analysis in the manuscripts to delve deeper into the effects of hyperparameter variations.
>
> **Question 3**: Furthermore, the ablation studies show that ME-Exp and w/o rank compare favourably with each other, even with XAL in some data sets.
>
> **Answer:** The comprehensive results, available in Appendix D.3, reveal the noteworthy performance of our model in comparison to other methods, particularly ME-Exp and w/o rank, across various tasks. It is evident that our model consistently outperforms these alternatives in most time. However, it's crucial to recognize that the impact of different components on the final performance may vary for each task, resulting in some fluctuations in performance across different scenarios.
>
> The harm of the ranking loss is not readily apparent in the results. Specifically, the model w/o rank, with a performance of 64.29%, exhibits a significant performance drop compared to XAL, which achieves a macro-F1 score of 67.16% in the task of Covid19. This discrepancy underscores the importance of the ranking loss in our model, as its absence significantly affects performance. These results highlight the significance of specific components in achieving optimal performance.
>
> **Question 4** : Interestingly, the authors have not conducted a comparative analysis of the computational costs associated with evaluating the acquisition functions of different active learning schemes. It would be of practical significance to assess the computational expenses involved in these methods.
>
> **Answer**: We conducted experiments to analyze the time consumption during each data query process in the MAMS task, which involves 7,090 training data instances. The results are as follows: ME-2 minutes, CA-2 minutes, BK-2 minutes, LC-2 minutes, BALD-11 minutes, Coreset-54 minutes, and our model XAL-21 minutes. Upon observation, it's apparent that our model requires more time for querying unlabeled data when compared to methods that leverage model uncertainty. However, it consumes less time than the representativeness-based method Coreset. We will incorporate these experimental findings into the manuscripts, detailing the time consumption across various tasks for a more comprehensive understanding of the performance characteristics of our model.
>
> [1] Deep batch active learning by diverse, uncertain gradient lower bounds
>
> [2] Cold-start active learning through self-supervised language modeling,
>
> [3] Diversity enhanced active learning with strictly proper scoring rules,”
>
> [4] ALICE: Active Learning with Contrastive Natural Language Explanations (Liang et al., EMNLP 2020)

---

### Official Review · Reviewer_C5EW · 2023-11-07

**Soundness:** 2 fair
**Presentation:** 2 fair
**Contribution:** 1 poor
**Rating:** 3
**Confidence:** 5

**Summary:**

This paper proposes an explanation-based active learning framework for text classification tasks. The framework selects the most informative samples for annotation, and generates and scores explanations for the classifier's predictions, then it forms a learning objective which combines the classification loss, the explanation generation loss, and the explanation ranking loss. To train the decoder for explanation generation, this paper leverages LLMs to obtain golden explanations. The proposed framework is evaluated on six text classification tasks in a low-resource setting, and the results show that it outperforms basic AL methods and coreset and contrastive AL.

**Strengths:**

1. This paper injects explanations to uncertainty-based AL process to prevent overconfidence and insufficient exploration.

2. The paper is presented in a coherent manner and easy to follow.

**Weaknesses:**

1. My major concern is whether generating explanation is the most efficient way to use LLMs in this paper's setting. Based on the examples in Figure 3, a budget of 500 instances corresponds to 1500 explanations. If we task LLMs to generate labels, instead of explanations, within the same API calling times, we could obtain annotated data that is N times greater, where N equals the number of labels (N=3 in this example). Such an increase in annotated data could markedly enhance model performance, particularly in low-resource Active Learning (AL) scenarios.

2. The first weakness also undermines the fairness of the comparison within the given data selection budget. The XAL model uses LLMs to produce high-quality explanations, incurring additional inference costs. A more compelling comparison would involve allocating an equivalent amount of LLM resources to the baseline methods for acquiring more labeled data.

3. The scope of datasets evaluated in this study is somewhat narrow. The number of classes is either 2 or 3, and the training sets do not exceed 8k instances.

4. This paper does not provide an analysis of the computational complexity of the proposed framework. While the paper mentions that the proposed framework requires more time and computational resources for training than encoder-only classifiers, it does not provide a detailed analysis of the computational requirements of the proposed framework.

**Questions:**

See above.

---

> ### Author Response · Authors · 2023-11-21
> **Response to the Weakness**
>
> We greatly appreciate the careful comments and suggestions provided by the reviewer. Our response to the weakness is shown as follows:
>
> **Question 1-2**: My major concern is whether generating explanation is the most efficient way to use LLMs in this paper's setting.  A more compelling comparison would involve allocating an equivalent amount of LLM resources to the baseline methods for acquiring more labeled data.
>
> **Answer**: In the manuscript, we detailed the performance of ChatGPT across various tasks, with macro-F1 scores ranging from 40% to 70%. However, calling the API to generate more labeled data comes with the caveat that annotation accuracy cannot be assured. Our attempts to conduct experiments in MAMS, following your suggestion, involved training the model in each active iteration. During this process, we utilized ChatGPT to annotate data randomly selected from D_u, which is three times the size of D_l. The obtained macro F1 scores were 51.08, 52.28, 53.79, 52.74, and 53.66 for data quantities ranging from 100 to 500 gold data, respectively. Despite these efforts, it became evident that incorporating additional labeled data from ChatGPT did not lead to an enhancement in model performance, primarily due to the lack of guaranteed accuracy in the annotation process. We will add the experiments in the manuscript and elaborate our description.
>
> **Question 3**: The scope of datasets evaluated in this study is somewhat narrow. The number of classes is either 2 or 3, and the training sets do not exceed 8k instances.
>
> **Answer**: In our experiments, we assessed the efficacy of our model across six classification tasks, encompassing a low-resource setting and varying difficulty levels. It's worth noting that while XAL demonstrated its utility in this context, its applicability extends beyond these specific tasks, making it adaptable to scenarios involving more label classes and diverse industrial downstream applications.
>
> The training set for MAMS consists of 7,000 instances, providing a solid foundation for expanding the model to accommodate additional classes and instances seamlessly. To enhance the comprehensiveness of our findings, we plan to incorporate a dataset with a greater number of classes and instances into the manuscript. This expansion will further highlight the versatility and scalability of our model across a broader spectrum of classification challenges.
>
> **Question 4**: This paper does not provide an analysis of the computational complexity of the proposed framework.
>
> **Answer**: We conducted experiments to analyze the time consumption during each data query process in the MAMS task, which involves 7,090 training data instances. The results are as follows: ME-2 minutes, CA-2 minutes, BK-2 minutes, LC-2 minutes, BALD-11 minutes, Coreset-54 minutes, and our model XAL-21 minutes. Upon observation, it's apparent that our model requires more time for querying unlabeled data when compared to methods that leverage model uncertainty. However, it consumes less time than the representativeness-based method Coreset. We will incorporate these experimental findings into the manuscripts, detailing the time consumption across various tasks for a more comprehensive understanding of the performance characteristics of our model.